# Toxic Y chromosome: Increased repeat expression and age-associated heterochromatin loss in male Drosophila with a young Y chromosome

**Alison H. Nguyen**, **Doris Bachtrog** *

Department of Integrative Biology, University of California Berkeley, Berkeley, California, United States of America

* dbachtrog@berkeley.edu

**Data Availability Statement:** All the sequencing data have been posted on GenBank under BioProject PRJNA644734.

## Abstract

Sex-specific differences in lifespan are prevalent across the tree of life and influenced by heteromorphic sex chromosomes. In species with XY sex chromosomes, females often out-live males. Males and females can differ in their overall repeat content due to the repetitive Y chromosome, and repeats on the Y might lower survival of the heterogametic sex (toxic Y effect). Here, we take advantage of the well-assembled young Y chromosome of *Drosophila miranda* to study the sex-specific dynamics of chromatin structure and repeat expression during aging in male and female flies. Male *D. miranda* have about twice as much repetitive DNA compared to females, and live shorter than females. Heterochromatin is crucial for silencing of repetitive elements, yet old *D. miranda* flies lose H3K9me3 modifications in their pericentromere, with heterochromatin loss being more severe during aging in males than females. Satellite DNA becomes de-repressed more rapidly in old vs. young male flies relative to females. In contrast to what is observed in *D. melanogaster*, we find that transposable elements (TEs) are expressed at higher levels in male *D. miranda* throughout their life. We show that epigenetic silencing via heterochromatin formation is ineffective on the TE-rich neo-Y chromosome, presumably due to active transcription of a large number of neo-Y linked genes, resulting in up-regulation of Y-linked TEs already in young males. This is consistent with an interaction between the evolutionary age of the Y chromosome and the genomic effects of aging. Our data support growing evidence that "toxic Y chromosomes" can diminish male fitness and a reduction in heterochromatin can contribute to sex-specific aging.

## Author summary

Y chromosomes can be toxic. The Y chromosome of many species contains a large number of transposable elements (TEs), which are transcriptionally constrained by repressive chromatin marks. When relieved of these epigenetic modifications, many TEs can readily move from one genomic location to another. We show that TEs located on the Y chromosome are less effectively silenced in male Drosophila, and the toxic Y effect appears more

---

**Funding:** This work was supported by NIH grants (nos. R01GM076007, R01GM101255 and R01AG057029) to DB. The funders had no role in study design, data collection and analysis, decision to publish, or preparation of the manuscript.

**Competing interests:** The authors have declared that no competing interests exist.

pronounced in a species that contains a larger Y chromosome with more repeats and more actively transcribed genes. Our data demonstrate that repeat-rich Y chromosomes are a genomic liability for males.

## Introduction

Males and females differ in many life history traits, and sexual dimorphism in fitness-related traits is often driven by natural selection [1]. However, some traits could vary between the sexes in a non-adaptive manner. For example, males and females often differ in their lifespan [2], and the chromosomal sex determination system has been shown to influence sex-specific longevity. In particular, several studies have suggested that the sex with the heteromorphic sex chromosomes (males in XY species; females in ZW species) has a shorter lifespan on average [3, 4]. The aging process is associated with an overall loss of heterochromatin in many species [5, 6], and differences in the heterochromatin content between sexes could in principle contribute to sex-specific mortality (the toxic Y effect; [7]). In species with heteromorphic sex chromosomes, the amount of repetitive DNA and thus heterochromatin can vary dramatically between males and females, due to the presence of a large, repetitive Y (or W) chromosome in the heterogametic sex. Heterochromatin loss can result in de-repression and mobilization of silenced transposable elements (TEs) [8–11], and might disproportionally affect the sex with the higher heterochromatin content.

We previously showed that heterochromatin loss differs between the sexes in *D. melanogaster* [12]. TEs are up-regulated in old male flies, and Y-linked repeats are especially prone to become de-repressed during aging in *D. melanogaster*. Genetic manipulations of sex chromosome karyotypes suggested that the Y chromosome directly contributes to sex-specific aging in flies: females containing a Y chromosome (XXY females) lived shorter than wildtype XX females, males lacking a Y (X0 males) outlived wildtype males, and males containing two Y chromosomes (XYY males) showed a drastically reduced lifespan compared to XY males [12]. However, repeat-rich Y chromosomes are typically poorly assembled (only 14.6 Mb of the roughly 40-Mb large Y chromosome of *D. melanogaster* is present in the latest assembly; [13]), making it impossible to directly observe chromatin changes on the heterochromatic Y chromosome and link them to expression changes of repeats using genomic approaches.

Here, we test for an association between heteromorphic sex chromosomes and sex-specific heterochromatin formation and silencing of repetitive DNA in a species with a well-assembled Y chromosome [14]. We assay chromatin and gene expression profiles in young and aged males and females of *D. miranda* (**Fig 1A**), a model system for studying the molecular basis of sex chromosome differentiation [14–19]. *D. miranda* has a recently evolved neo-sex chromosome that was formed only 1.5MY ago from an ordinary autosome that fused to the ancestral Y [20]. The neo-X and neo-Y are in the process of differentiating into heteromorphic sex chromosomes, and the neo-Y has dramatically increased in size mainly through an accumulation of TEs [14, 15, 17, 21]. Specifically, a high-quality genome assembly that includes large fractions of heterochromatin contained >90 Mb of neo-Y-linked sequence (compared to ~25 Mb of neo-X linked sequence), and 65 Mb are derived from repetitive elements (compared to 5 Mb on the neo-X) [14]. TEs are uniformly enriched along the neo-Y chromosome and a main contributor to its dramatically increased genome size [14, 15, 17, 21]. For example, the most abundant repeat on the neo-Y is the *ISY* element, a helitron transposon that is inserted about 22,000 times on the neo-Y/Y chromosome and amounts to more than 16 Mb of DNA (i.e., 16% of neo-Y sequence), while its former homolog, the neo-X chromosome, only harbors about 1,500 copies (3% of the neo-X; less than 1 Mb; [14]). The second most common repeat class that has amplified on the neo-Y are *gypsy* TEs; roughly 14,300 insertions were found on

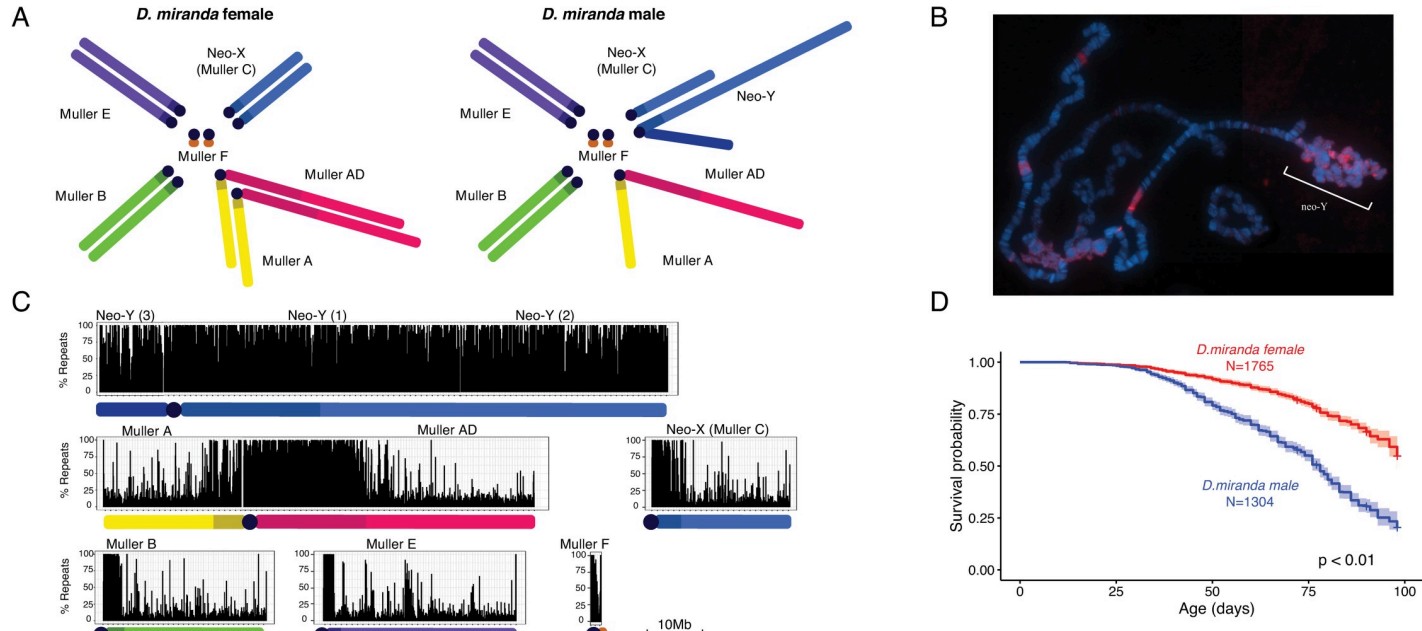

**Fig 1. *Drosophila miranda* karyotype and lifespan. A.** Karyotype of *D. miranda* males and females. Chromosome arms (Muller elements) are color-coded. **B.** Polytene chromosome squash of *D. miranda* male stained with *HP1* (magenta) marking heterochromatin. **C.** Genome-wide repetitive content per chromosome. Shown is repeat content (% repeat masked bp) in 50kb windows. **D.** Kaplan-Meier survivorship curves [72] for MSH22 males (blue) and females (red), with the shaded region indicating the upper and lower 95% confidence interval calculated from the Kaplan-Meier curves. The number of flies counted for each sex (n) to obtain the survivorship curves is indicated.

the neo-Y (15% of neo-Y sequence, 15 Mb) and less than 1 Mb on the neo-X (about 800 insertions, i.e., 3% of its sequence; [14]). Thus, a huge number of TEs has accumulated on the neo-Y chromosome since its origination 1.5 MY ago. Yet this chromosome still contains thousands of protein-coding genes that are embedded within these islands of TEs [14], and many of the neo-Y linked protein-coding genes are still actively transcribed [18, 22]. Thus, unlike the Y chromosome of *D. melanogaster*, the neo-Y of *D. miranda* contains thousands of transcribed, euchromatic genes, which are intermingled with repetitive heterochromatin.

We use chromatin and transcription profiling to show that heterochromatin formation is compromised on the repeat-rich neo-Y of *D. miranda* already in young males, resulting in global up-regulation of TEs in males. Age-associated heterochromatin loss is more pronounced in male flies compared to females, which also live shorter, and accompanied by de-repression of satellite repeats in old males. Our data provide empirical support that toxic Y chromosomes can diminish male fitness, and contribute to sex-specific aging in species with heteromorphic sex chromosomes.

## Results

### *Drosophila miranda* males have more repeats and live shorter

We chose the standard lab strain MSH22 of *D. miranda*, a fly that has served as a model for Y chromosome evolution (**Fig 1A**) [14, 18, 22] to investigate how the 'toxic' Y chromosome may influence sex-specific behaviors of repetitive sequences, such as TE expression, the dynamics of heterochromatin during the lifespan of a fly, and sex-specific aging. In *D. miranda*, an autosome fused to the ancestral sex chromosome about 1.5MY ago [20], creating a neo-Y chromosome which is at an intermediate stage of degeneration. The neo-Y still contains many of its

**Table 1. Assembled length for each chromosome, size of chromosome arms and pericentromere, and repeat-masked bases (in Mb).** The total genome size inferred from flow cytometry is given in brackets.

| | Total Assembly (Mb) | Euchromatic Arms (Mb) | Pericentric Heterochromatin | Repeatmasked (Mb) |
|---|---|---|---|---|
| X chr (A/D) | 77.6 | 57.1 | 20.5 | 28.9 |
| neo-X (C) | 25.3 | 21.9 | 3.4 | 6.1 |
| Y/neo-Y chr | 92.1 | n/a | n/a | 76.8 |
| chr4 (B) | 32.5 | 29.1 | 3.4 | 6.6 |
| chr2 (E) | 35.3 | 33.3 | 2 | 6.2 |
| dot | 2.4 | 0 | 2.4 | 1.3 |
| total | 173.1 | 141.4 | 31.7 | 49.1 |
| male (2n) | 335.4 (354) | 203.8 | 131.6 | 140.0 |
| female (2n) | 346.3 (351.2) | 282.9 | 63.4 | 98.2 |

ancestral genes, but has expanded in size by about 3-fold, mostly due to the accumulation of TEs [14, 17]. This increase in repeat content is accompanied by a change in its chromatin structure with many regions along the neo-Y becoming heterochromatic (**Fig 1B**) [23]. *D. miranda* has a high-quality genome assembly that comprise large stretches of repetitive heterochromatin, including major parts of the pericentromeres and the Y chromosome [14] (**Fig 1C**). In particular, *D. miranda* males have a enormous, ~92 Mb large recently formed and highly repetitive and heterochromatic neo-Y chromosome [14], while the pericentromeric heterochromatin on the X/neo-X only amounts to ~24 Mb (**Table 1**) [14, 24]. Thus, males contain significantly more repetitive DNA than females, and flow cytometry estimates agree with our assembled genome sizes (**Table 1**).

In most Drosophila species male flies live significantly shorter than female flies [25, 26]. We determined longevity for males and females for the reference lab strain from *D. miranda* (MSH22) at 18°C (from 3 biological replicates, see Materials & Methods). Median survivorship for males is 78 days, and >98 days for females (**Fig 1D**). Thus, lifespan assays confirm that males live significantly shorter than females, consistent with multiple studies on sex-specific lifespan in Drosophila [25, 26]. Longevity patterns are thus consistent with the notion that the repeat-rich Y chromosome reduces survivorship in the heterogametic sex [3, 4], but note that other factors besides sex (such as male-male aggression) may contribute to differences in longevity between males and females [27].

## Heterochromatin loss is more pronounced in males

We collected replicate ChIP-seq data (4 replica for each sex and age) for a histone modification typical of repressive heterochromatin (H3K9me3) from young (9–11 days) and old (80-90-day) *D. miranda* male and female brains (MSH22) (**S1 and S2** Tables). In order to increase mappability of our ChIP-seq data (both to repetitive regions, and to distinguish between reads derived from the neo-X and neo-Y), we collected 100-bp paired-end read data. We spiked-in a fixed amount of chromatin from *D. melanogaster* to each *D. miranda* chromatin sample prior to ChIP and sequencing, to compare the genomic distribution of chromatin marks across samples using a 'spike in' normalization method [12, 28]. This species pair is sufficiently diverged from each other that there is very little ambiguity in the assignment of reads to the correct species, especially for fast-evolving repetitive DNA (**S1 Appendix**), and this strategy allows us to compare levels of H3K9me3 enrichment across sexes and time.

**Fig 2** shows the genomic distribution of the repressive histone modification H3K9me3 for young and old male and female flies. As expected, heterochromatin is enriched at repetitive regions, including pericentromeres, the small dot chromosome and the repeat-rich Y

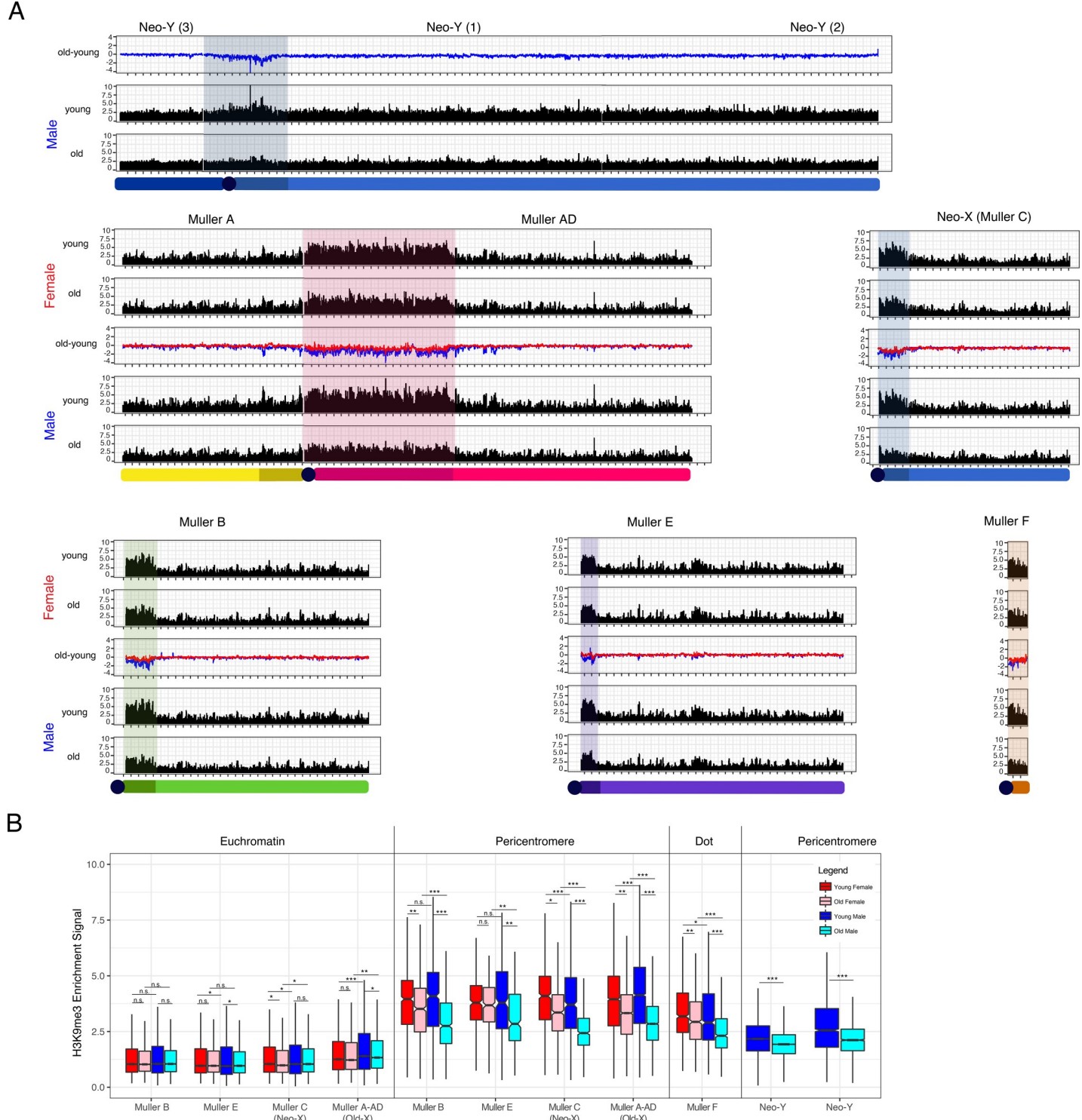

**Fig 2. Heterochromatin loss and aging. A.** Genome-wide enrichment of H3K9me3 for young and old *D. miranda* males and females along the different chromosome arms. Values represent the mean of 4 biological replicates for Young/Old Females and Young Males and the mean of 3 biological replicates for Old Males in 25kb windows. Subtraction plots show the absolute difference in signal of 25kb windows between young and aged flies along the chromosome arms, with females in red and males in blue. One tick mark is equal to 1 Mb on the x-axis. **B.** Boxplot showing ChIP enrichment differences between biological samples at euchromatin and pericentromeres. Enrichment resolution is in 5kb windows. Significance values calculated (* p<0.05, ** p<1e-6, *** p<1e-12, Wilcoxon test) for males (blue, cyan) and females (red, pink), with pericentromere boundaries defined by [24].

chromosome (**Fig 2A**). Overall heterochromatin enrichment is similar for repetitive regions at the X and autosomes in young males and females (**Fig 2B**). Consistent with heterochromatin being lost during aging, both males and females show a reduction in H3K9me3 enrichment at their pericentromeres in old compared to young flies (**Figs 2** and **3**). Strikingly, heterochromatin loss is more pronounced in old male flies than in old females (**Fig 2A and 2B**). This finding is reproducible across individual replicates (**S1**–**S6** Figs **and S3 Table**), and robust with regards to normalization strategy (**S1 Fig and S4 Table**) and using only uniquely mapping reads (**S3 Fig and S5 Table**). *D. miranda* males show significantly more regions that lose H3K9me3 signal (1.5-fold or more in 5-kb windows) during aging compared to females (2936 vs. 316, p < 0.00001; Fisher's exact test; **S7 Fig**). Almost all regions (78% of windows) that lose heterochromatin in males are located within the pericentromeres of the autosome/X (1963 windows) and the neo-Y (328 windows) (**S7**–**S9** **Figs)**. In females, 25.6% of windows (81) losing heterochromatin occur within the pericentromeres. Many fewer regions gain H3K9me3 signal (1.5-fold or more) during aging (947 in males and 155 in females, p < 0.00001, Fisher's exact test; **S7 Fig**). Almost all regions that gain H3K9me3 signal are found along the chromosome arms (**S7 Fig**), consistent with the idea that some heterochromatin becomes re-distributed in old flies [29]. Faster heterochromatin loss in old males is in agreement with recent findings in *D. melanogaster* [12]. Note that genes involved in heterochromatin formation and organization show little change in their expression during aging in both males and females, suggesting that heterochromatin loss in males is not due to a male-specific decline in proteins required for heterochromatin formation (**S6 Table**). These data suggest that increased deterioration of heterochromatin during aging may be a general feature of the sex with more repetitive DNA.

## Heterochromatin content and loss on the neo-Y chromosome

The neo-Y is highly repetitive (**Fig 1** **and** **Table 1**) and enriched for heterochromatin (**Fig 2**), yet to a lesser extent than pericentromeric regions (**Fig 2A and 2B**). Note that while the neo-X and neo-Y still show considerable homology at protein-coding genes, there is little mis-mapping between reads derived from the neo-X or neo-Y chromosome (**S2 Appendix**), indicating that lower H3K9me3 enrichment on the neo-Y is not an artifact due to mis-mapping. The neo-Y harbors several thousand protein-coding genes embedded in highly repeat-rich regions [14, 22], and overall repeat content on the neo-Y is slightly less than of pericentromeres on the X or autosomes (72% of bp are repeat-masked on the neo-Y chromosome, compared to 82–89% of the pericentromeres on X and autosomes).

To evaluate if lower repeat content on the neo-Y can account for lower levels of heterochromatin, we compared H3K9me3 enrichment versus repeat content for 5-kb windows across chromosomes (**Fig 3A**). As expected, genomic regions with a higher repeat density show more heterochromatin, and absolute levels of H3K9me3 enrichment are very similar between the X and autosomes, and young males and females. However, for repeat-rich windows (>70% repeat masked), the neo-Y chromosome shows considerably lower levels of H3K9me3 enrichment than other chromosomes, even in young males (**Fig 3A**). In particular, average H3K9me3 enrichment in highly repetitive windows (>70% of bases repeat masked) is about 30% lower on the neo-Y compared to X-linked or autosomal regions in young males, and 24% lower in old males. The same conclusion holds if the top repeat-rich regions (>99.5% repeat masked) are considered (**Fig 3A**). Thus, this suggests that repetitive DNA on the neo-Y shows less heterochromatin-induced epigenetic silencing than other repeat-rich regions of the genome. Immunofluorescence staining of H3K9me3 in mitotic chromosomes confirms much weaker labeling of the neo-Y chromosome relative to the large pericentromeric region of the X chromosome in male larvae (**Figs 3B and S10**). While we observe consistent and robust

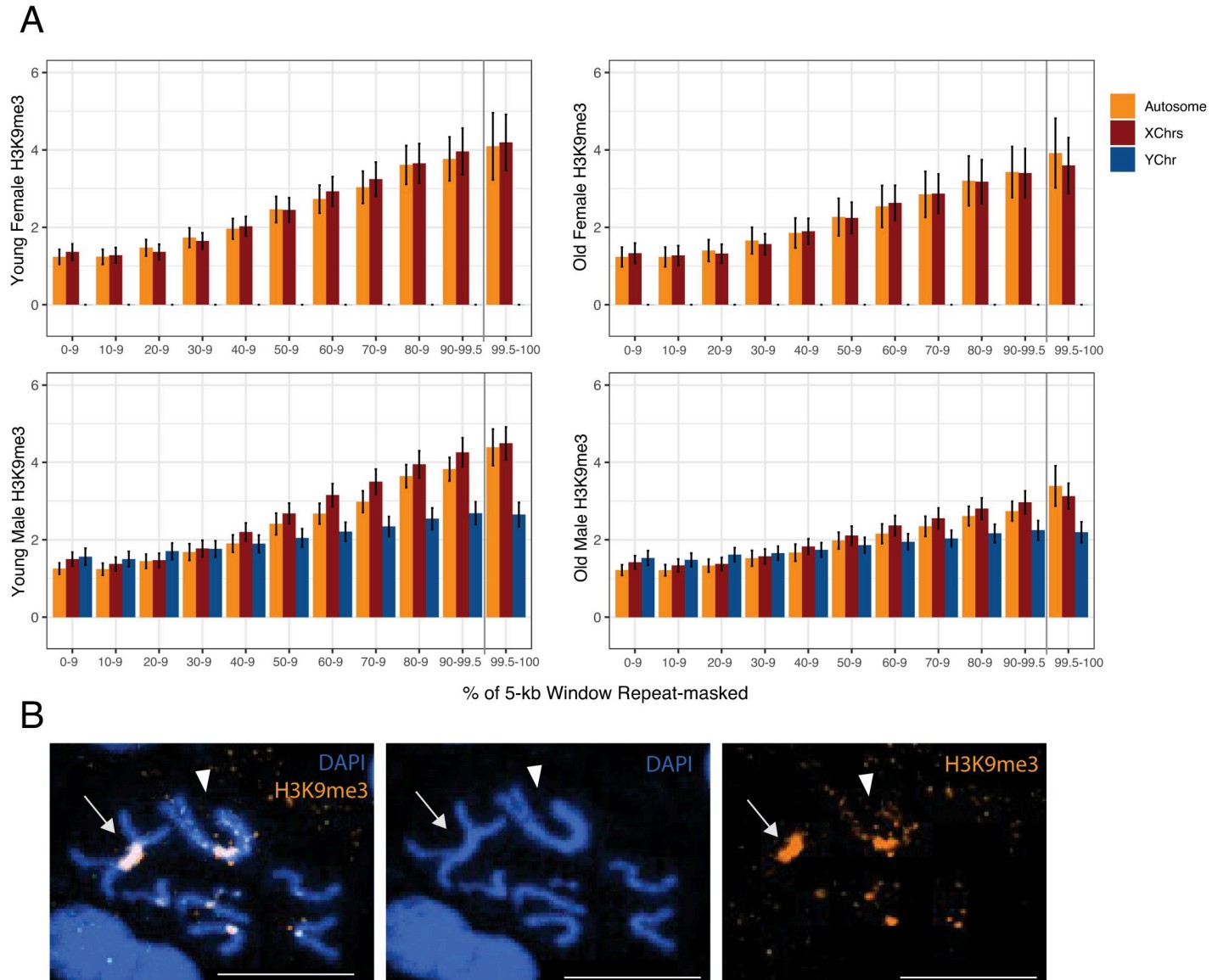

**Fig 3. Heterochromatin enrichment across chromosomes. A.** ChIP enrichment across samples grouped by chromosome and binned by amount of repeats (%) per 5kb–window. Values represent the mean of 4 replicates with standard error bars with the exception of the "Old Males" where we omit 1 failed ChIP replicate. **B.** Immunofluorescence staining for H3K9me3 in male mitotic chromosomes. Arrowhead denotes neo-Y, arrow denotes MullerA-AD (old-X), scale bar is 50μm.

H3K9me3 staining along the repetitive pericentromere on the X chromosome and neo-X/ autosomes, only the pericentromeric region of the neo-Y shows intense H3K9me3 staining; the highly repetitive chromosome arms of the neo-Y show considerable weaker H3K9me3 signal (**Fig 3B**), consistent with our ChIP-seq results (**Fig 2**). Thus, Immunofluorescence staining independently confirms that the neo-Y chromosome contains lower levels of heterochromatin, compared to other repetitive regions in the genome.

Multiple factors could contribute to lower levels of H3K9me3 enrichment on the neo-Y. The neo-Y chromosome still contains many unique regions with transcribed protein-coding genes [14, 18, 22] and thus encompasses a patchwork of heterochromatic repeats interspersed with unique and genetically active euchromatic regions. In Drosophila, the boundary between

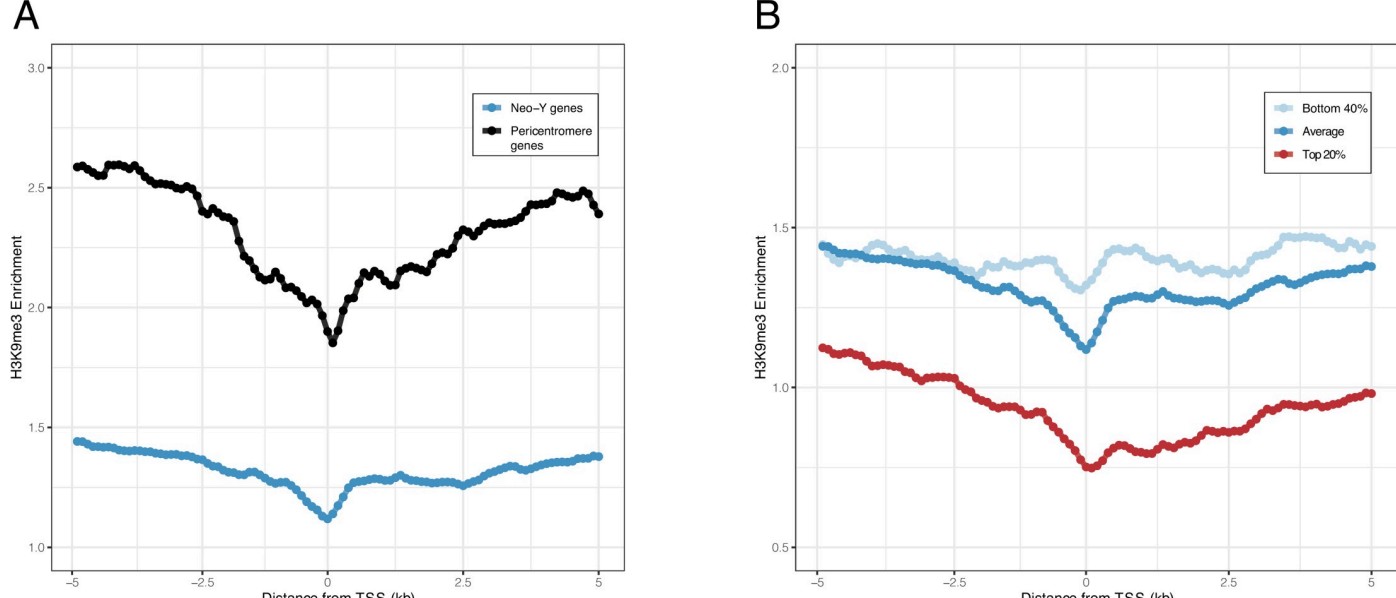

**Fig 4. Heterochromatin enrichment at transcription start sites (TSS) on the neo-Y and the pericentromeres. A.** The average young male H3K9me3 ChIP enrichment profile +/- 5kb of TSS in 100bp resolution grouped by gene location (neo-Y or pericentromere). **B.** The average young male enrichment profile at TSS neo-Y genes grouped and plotted by expression level. The resolution is also 100bp.

heterochromatin and euchromatin is not fixed but determined by the local balance between factors that promote either heterochromatin or euchromatin [30]. Active transcription could limit spreading of heterochromatin on the neo-Y and reduce overall H3K9me3 enrichment [31]. Indeed, the gene density on the neo-Y is substantially higher than in the pericentromeres. The *D. miranda* annotation contains 255 genes located in the pericentromeres (that is, roughly 9 genes/Mb) and 5625 genes on the Y/neo-Y (61 genes/Mb) [14]. Most genes are transcribed in adult brain samples (246 of all the pericentromeric genes and 4428 of the neo-Y genes; see below), consistent with the idea that active transcription of neo-Y genes could impede heterochromatin formation on this chromosome. Indeed, we find that H3K9me3 enrichment is lower around transcription start sites (TSS) for genes located on the neo-Y, similar to genes located in pericentromeric regions (**Fig 4A**), supporting the idea that active transcription counteracts local heterochromatin formation. Importantly, if we group neo-Y linked genes into those that are highly expressed from the neo-Y (top 20%) versus those lowly expressed (bottom 40%), we find that H3K9me3 enrichment is substantially lower neighboring genes that are highly expressed (**Fig 4B**). Lower H3K9me3 levels near highly transcribed genes supports the notion that active transcription interferes with heterochromatin formation on the neo-Y.

In addition, heterochromatin on the neo-Y is evolutionarily young, and its overall repeat structure may differ from pericentromeres. The neo-Y was an ordinary autosome until 1.5 MY ago and only acquired its high repeat content and heterochromatin structure after it became fused to the ancestral Y [15, 21]. The repeats present on the neo-Y should therefore encompass mostly recently active TEs, and it is possible that these repeats may initiate less heterochromatin [32]. Indeed, the relative abundance of different TE families differs between the neo-Y chromosome and the rest of the genome (**S11 Fig**; [14]); we find the neo-Y contains more LTR transposons and helitrons but less simple repeats in comparison to other chromosomes (**S11 Fig**).

While overall TE composition differs between the neo-Y and other chromosomes, transposons belonging to the same family may also show lower levels of H3K9me3 enrichment if

present on the neo-Y. As expected, H3K9me3 profiles at TE families show that there is a general enrichment of this repressive mark in both male and female flies ([Fig 5A]). To test if neo-Y linked TEs show less H3K9me3 enrichment, we calculated average H3K9me3 enrichment for

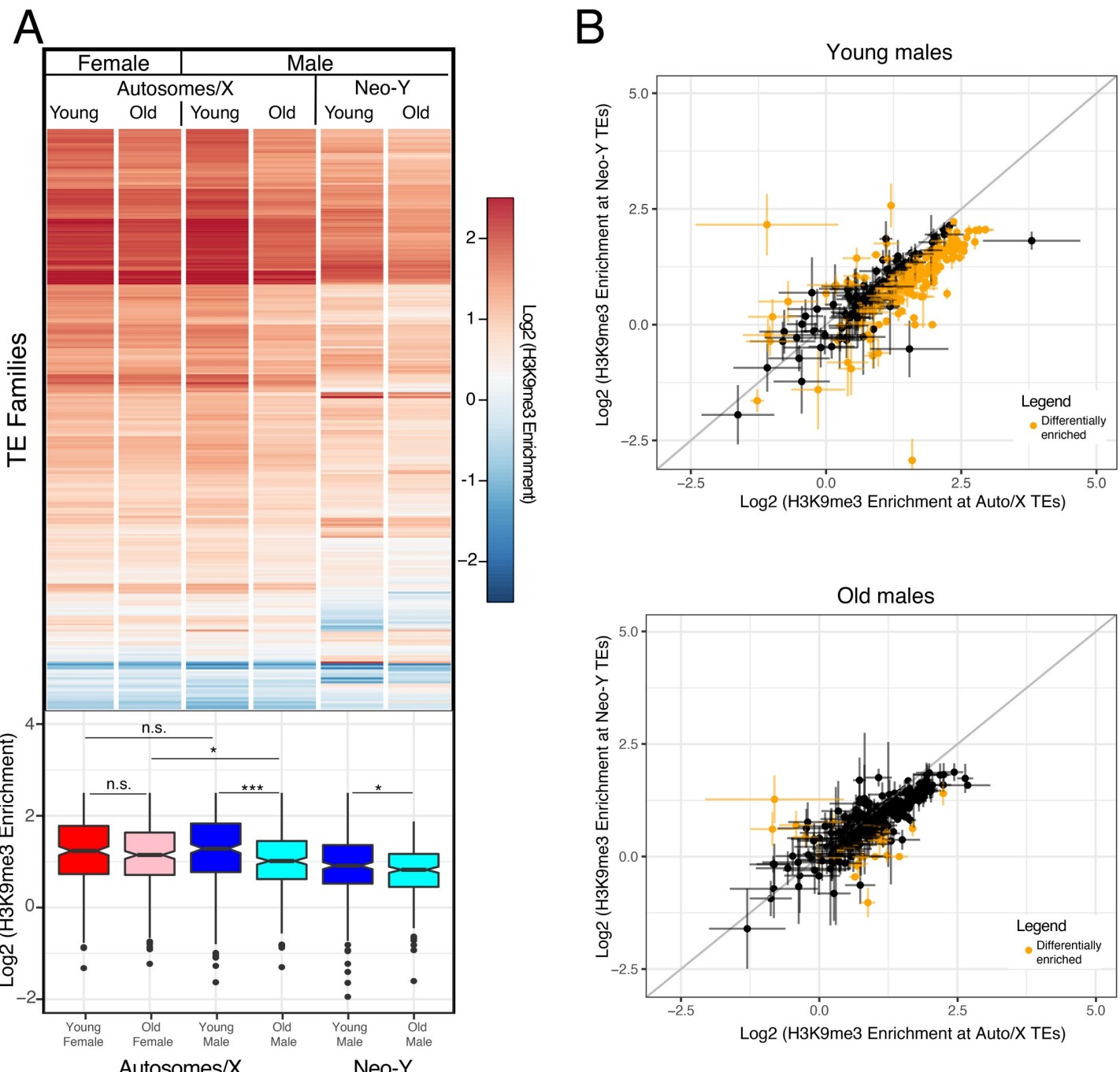

**Fig 5. Heterochromatin enrichment at TEs across sex, time and chromosomes.** Shown is enrichment of H3K9me3 at different repeat families for young and old females and males. **A.** Log2 H3K9me3 enrichment of all transposable elements (from [55]) in young (5–9 day) and old (80–98 day) male and female MSH22, averaged across 4 replicates for Young/Old Females and Young Males and across 3 replicates for Old Males. H3K9me3 enrichment in males is separated by autosomes/X and neo-Y. Boxplots below with significance values calculated using the Wilcoxon test (* p< 0.05, ** p<0.01, *** p<1e-5). **B.** Log2(H3K9me3 enrichment) differences between TEs found on the Autosomes/X and the neo-Y for young (top) and old (bottom) males. Data represents averages of 4 replicates for Young/Old Females and Young Males and of 3 replicates for Old Males. The bars denote standard error. Data highlighted in yellow are TEs that show significant differences in H3K9me3 enrichment on autosomes/X and neo-Y (p < 0.05, two-tailed Student's t-test).

each TE family based on genome-wide mapping of H3K9me3 separately for copies located on the neo-Y versus other genomic locations. Indeed, copies of TEs inserted onto the neo-Y typically show lower levels of H3K9me3 enrichment than copies on other chromosomes. Global H3K9me3 enrichment is about 30% lower for neo-Y TEs compared to their X- and autosomal homologs in young flies (and 27% lower in old flies), and 92/16 show significantly less/more H3K9me3 if located on the neo-Y relative to copies on the X/autosome in young males (and 25/6 in old ones; **Fig 5B**). Thus, this suggests that most TEs experience less epigenetic silencing on the neo-Y relative to other regions of the genome.

Like other repeat-rich regions, overall levels of heterochromatin on the neo-Y decrease during aging; however, the absolute amount of H3K9me3 loss appears less on the Y in old males, compared to pericentric regions, or the dot chromosome (**Figs 2** and **S12**). Overall, our results show that male *D. miranda* lose heterochromatin marks more rapidly than female, consistent with our previous findings in *D. melanogaster* [12]. However, we also find that the neo-Y shows considerably lower levels of heterochromatin than what we might expect given its repeat content. Indeed, levels of H3K9me3 enrichment on the neo-Y in young males resemble reduced levels of heterochromatin at X-linked and autosomal pericentromeres in old males.

## Repeat de-repression in young male flies

Heterochromatin silences repetitive DNA [31], and reduced levels of H3K9me3 along the large repeat-rich neo-Y chromosome could have profound consequences on TE silencing in male *D. miranda* even in young flies. To study TE expression changes between sexes and during aging, we gathered replicated stranded RNA-seq data from young and old *D. miranda* (3 biological replicates, 100-bp paired-end reads to increase mappability; **S7**–**S9 Tables**). Indeed, comparison of TE expression between young and old male and female *D. miranda* reveals that repeat expression is strikingly increased in male flies compared to females, irrespective of age (**Figs 6A** and **6B** and **S13**). In particular, 135 TE families are significantly upregulated, while 1 is expressed at lower levels in male *D. miranda* compared to females already in 9-day old flies, and the fraction of transcripts derived from TEs is more than twice as high in males compared to females (the fraction of repetitive reads in all RNA-seq reads is on average 8.0% at young males and 3.7% at young females, **S8**–**S10 Tables**).

It is difficult to identify the genomic copy from which a TE transcript originated, especially for recently active families with highly similar copies. We used several approaches to test if increased TE expression in males is primarily driven by repeats located on the neo-Y. First, we mapped RNA seq reads to the *D. miranda* genome, and estimated expression of neo-Y linked copies versus copies on other chromosomes for each TE family [33]. We find that 53% of all TE transcripts in young males are derived from the autosomes and X and 47% are from the neo-Y, suggesting that most of the excess TE transcription in males could be attributed to expression from the neo-Y chromosome (**S14** and **S15 Figs**).

We also quantify expression of TEs enriched on the neo-Y independent of our assembly (**Fig 6A and 6B**). TE families with more copies on the neo-Y chromosome will show higher coverage in male DNA sequencing libraries compared to female libraries [12]. We ranked TE families based on their male/female genomic coverage and classified 81 TEs as Y-enriched (based on log2(female/male gDNA) coverage; Wei et al. 2020). Indeed, TE families that are enriched on the neo-Y show higher expression in both males and females than the remaining TE families that are not enriched on the neo-Y (**Fig 6D**). In addition, most of the neo-Y enriched TE families are significantly more expressed in males, with 85%

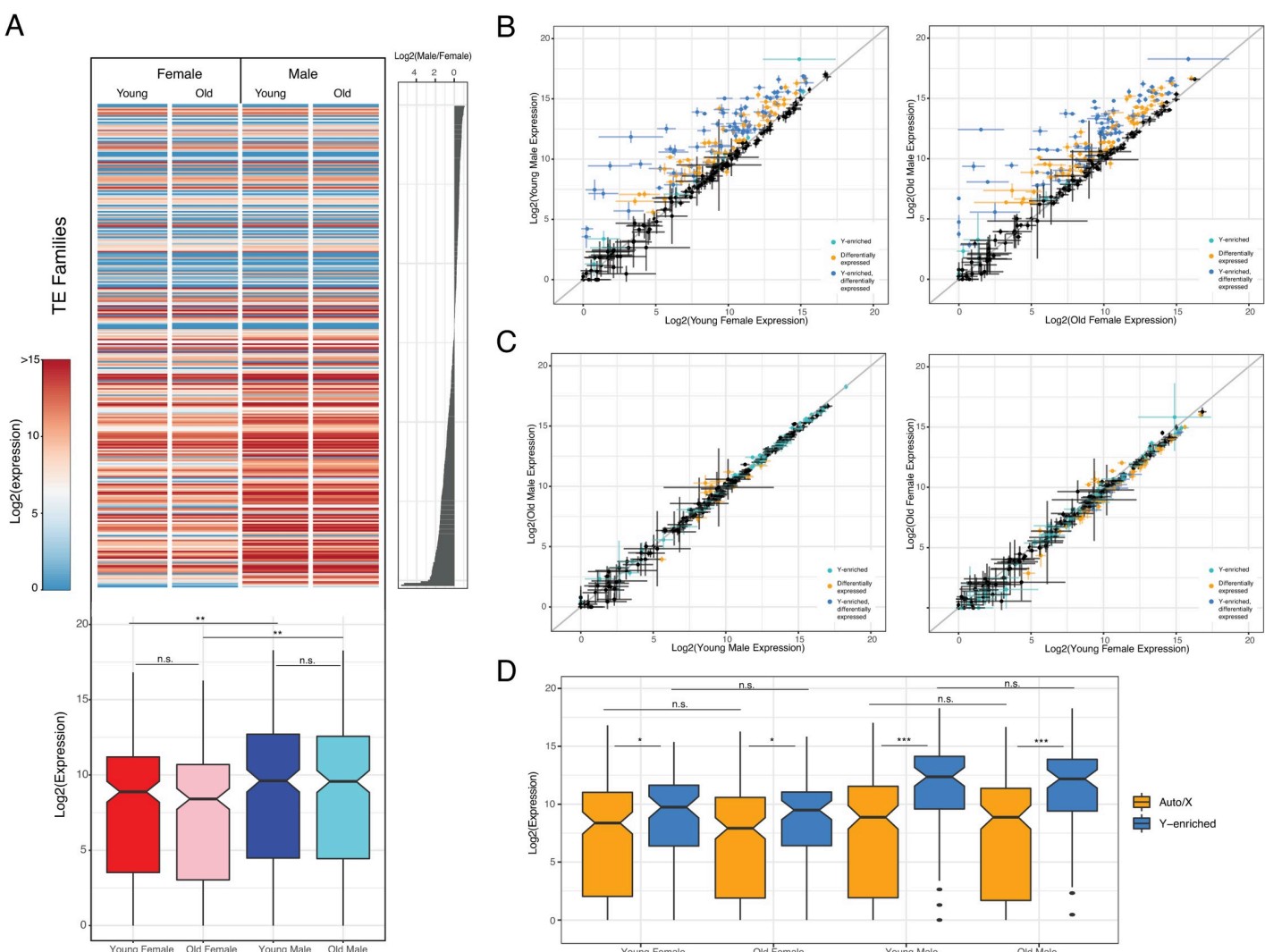

**Fig 6. TEs are upregulated in male *D. miranda*.** Shown is differential TE expression within and between sexes. **A.** Overall log2 expression by sex and age with rows sorted by log2 female/male TE coverage (right histogram). Boxplots below with significance values calculated using the Wilcoxon test (* p< 0.05, ** p<0.01, *** p<1e-5). Data represents averaged expression across samples. **B.** Log2 expression differences of TEs between young (left) and old (right) males versus females. Data indicates mean value of 3 biological replicates and bars denote standard error. Color-code indicates significant comparisons (50% significant higher or lower expression; Wald test, p <0.05) and Y-enriched repeats (based on genomic coverage): cyan: Y-enriched, p <0.05; yellow: unbiased, p <0.05; blue: Y-enriched, n.s.; black: unbiased; n.s. Grey line indicates 1:1 expression levels. **C.** Same as B but comparison between young vs. old males (left) and females (right). **D.** Log2 TE expression by sex and age grouped by TEs enriched on the neo-Y (blue) and TEs not enriched on the neo-Y (yellow). Significance values calculated using the Wilcoxon test (* p< 0.05, ** p<0.01, *** p<1e-5).

of the neo-Y enriched TEs (69 families) being more highly expressed in young males compared to females. Thus, this confirms the expectation that it is more recently active TEs that invaded the neo-Y chromosome since it stopped recombining with the neo-X 1.5MY ago. Increased copy number of these TEs and lower levels of heterochromatin on the neo-Y might drive their higher expression in males. Note that increased TE expression in male holds even if we account for higher copy number of repeats in males by normalizing by genomic DNA read counts or by estimated copy numbers in the genome assembly (**S16 Fig**). Thus, higher repeat expression in males is not simply a product of higher TE copy number in males.

## No change in TE expression in old flies

A loss of heterochromatin in old individuals can lead to a re-activation of previously silenced TEs [8–11]. In *D. melanogaster*, we found that TEs are efficiently repressed in young male and female brain tissue [12]. While old females maintained heterochromatin and silencing of repeats, males aged for a similar amount of time showed a dramatic reduction in heterochromatin at pericentric regions and on their Y chromosome. Heterochromatin loss in males was accompanied by de-repression of TEs, and repeats from the Y chromosome were found to be disproportionally up-regulated in old *D. melanogaster* males [12].

Female *D. miranda* efficiently suppress TEs throughout their life, similar to *D. melanogaster*. In females, 4 TEs showed increased and 36 decreased expression during aging (**Fig 6A**, using abs(log2(fold-change)) = 0.58), and the total fraction of transcripts derived from repeats decreases during aging (the fraction of repetitive reads in all RNA-seq reads is on average 3.7% at 9 days, vs. 2.9% at 80 days, **S3 Table**). In contrast to repeat de-repression found in old *D. melanogaster* males, however, expression of TEs does not change during aging in male *D. miranda* (**Figs 6C and S13**). In particular, 6/8 TEs show a in/decrease in expression during aging in *D. miranda* males (**Figs 6C and S13**; same fold-change as females), and the total fraction of transcripts derived from repeats slightly decreases during aging (from 8.0% at 9 days to 7.6% at 80 days, **S3 Table**). Thus, while old *D. miranda* males show a drastic loss in heterochromatin at pericentromeres on the X and autosomes, this does not result in global expression changes at TEs. However, unlike in *D. melanogaster*, overall levels of TE expression are substantially increased in male *D. miranda* relative to females, already in young flies (**Figs 6B and S13**).

## Satellite mis-expression in old male flies

Another type of repetitive DNA that is targeted by heterochromatin is satellite DNA repeats [31]. Satellites are often transcribed, and satellite transcripts may have important biological roles, including heterochromatin formation or centromere assembly [34–37]. Unlike *D. melanogaster*, which has large arrays of satellite DNA composed of simple repeats (sometimes several megabases in size), the genomes of *D. miranda* and its close relatives contain many fewer, often shorter satellites [24, 38]. Of the 780 different satellite repeat units that were identified in the *D. miranda* genome assembly [14], only 9 are enriched on the neo-Y (based on male/female genomic coverage), compared to 81 neo-Y/ Y enriched TEs (out of 303 TE families total; see above). Thus, unlike TE's, satellites are not preferentially found on the neo-Y. Satellite DNA is highly enriched for H3K9me3 in young flies (**Fig 7A**), in both males and females. Mimicking overall patterns of heterochromatin enrichment, we see a clear drop in H3K9me3 levels in old male flies at satellites, but not females (**Fig 7A**).

Many satellite repeats are transcribed, and at similar levels in both sexes; 21/16 satellites show higher/lower expression in young males compared to young females (**Figs 7B and S17**). Thus, unlike TEs that are highly enriched on the neo-Y and inefficiently silenced in males (**Fig 6**), no such differences in satellite expression are seen in young males and females. Old males show a dramatic loss in repressive heterochromatin at satellites, and a simultaneous up-regulation at satellite DNA expression. In males, 21 satellites show increased expression during aging compared to 1 satellite showing decreased expression, while expression at 2/1 satellites in/decreased in old females (**Figs 7C and S17**). Thus, a loss of silencing heterochromatin at satellites in old male *D. miranda* is accompanied by a de-repression of satellite transcripts.

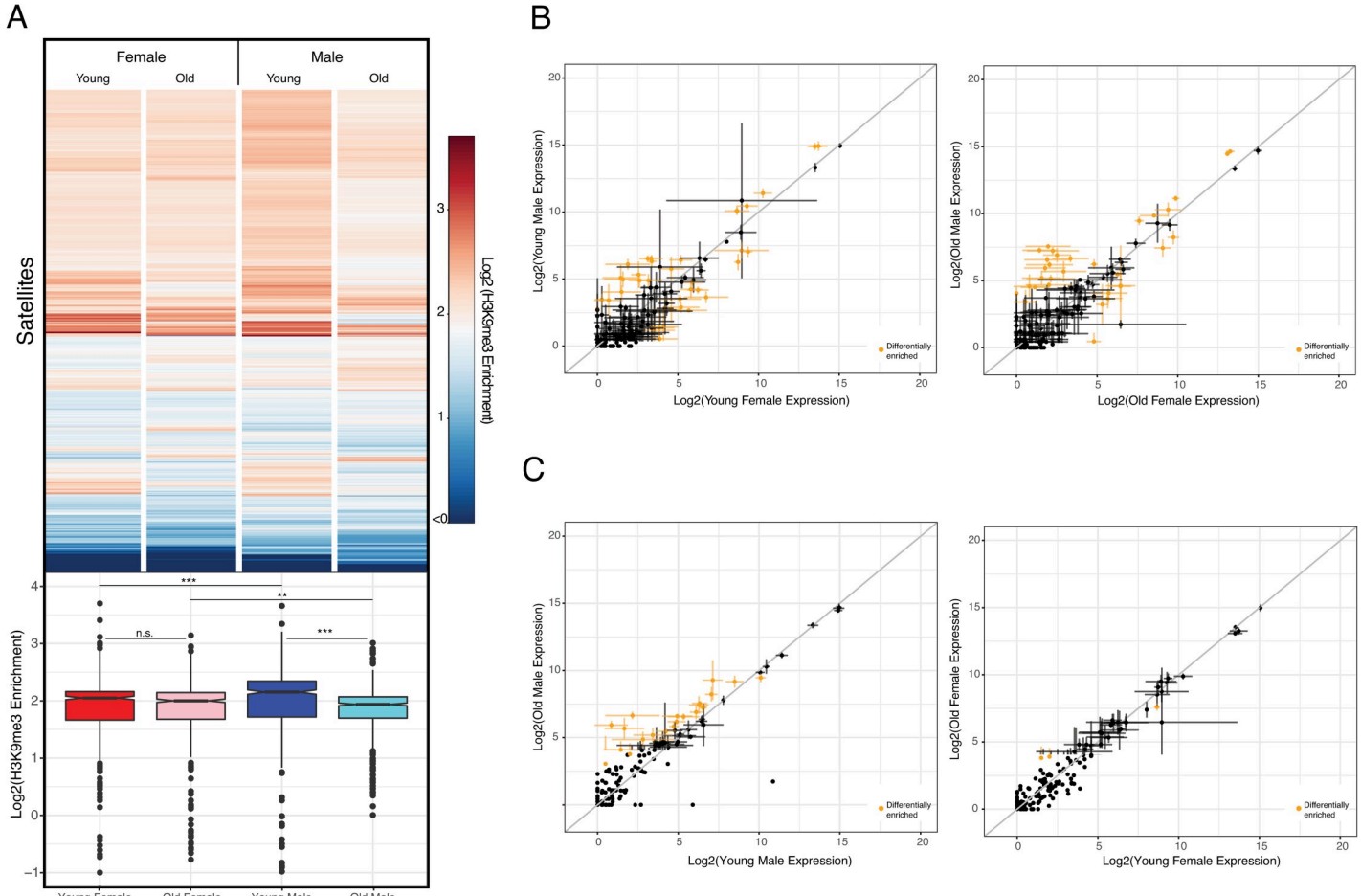

**Fig 7. Heterochromatin and expression at satellite DNA. A.** Average log2 H3K9me3 enrichment at different satellite DNA for young and old females and males. Boxplots with significance values calculated using the Wilcoxon test (* p< 0.05, ** p<0.01, *** p<1e-5). **B.** Log2 expression differences between young (left) and old (right) males and females. Data represents mean value of 3 replicates with bars denoting standard error, data in yellow denote significant differentially expressed satellites (50% higher or lower expression, Wald test, p < 0.05). Grey line denotes 1:1 expression levels between the samples. **C.** Log2 expression differences between young vs. old males (left) and females (right). Plots made in similar fashion as in **B**.

## Mis-expression of neo-Y genes and genes embedded in heterochromatin in old flies

Changes in heterochromatin during aging could also affect protein-coding genes; while most genes are located in repeat-poor euchromatin, some are embedded in heterochromatin. In addition to protein-coding genes located on the neo-Y (5586 in our combined ChIP and RNA analysis), *D. miranda* also has a total of 376 genes in pericentromeres and the dot chromosome. Genes that are naturally located in heterochromatin, such as pericentric genes in Drosophila or genes on the dot chromosome, have evolved mechanisms to assure gene transcription in this normally suppressive chromatin environment, and often are dependent on it [39–41]. Indeed, genes located on the pericentromeres appear to be expressed at slightly lower levels in old flies, compared to young ones, in both sexes (6/18 significantly up/downregulated in aged males and 3/29 in aged females; S18 Fig). Neo-Y genes, on the other hand, only became embedded in their heterochromatic environment within the past 1.5MY. Indeed, as flies age and lose H3K9me3, neo-Y genes become slightly upregulated (249/141 are significantly up/downregulated; S18 Fig). Thus, these results are consistent with a dynamic view of

co-evolution of heterochromatin and its host genome; genes embedded in old heterochromatin, such as the pericentromere, had millions of years of evolution to properly function in this typically repressive environment, and depend on it. A loss of heterochromatin during aging thus results in downregulation of pericentric genes. In contrast, genes that only became invaded by repeats and thus heterochromatin more recently, such as neo-Y genes in *D. miranda*, have not yet had enough time to adopt to this novel environment; losing heterochromatin, therefore, results in slight upregulation of neo-Y genes.

## Discussion

Repetitive DNA is composed of TEs and satellite DNA, and their misregulation can induce genomic instability [42]. Transposable elements (TEs) are mobile DNA sequences that can change their position and comprise a substantial fraction of eukaryotic genomes [43]. Satellite DNA is often found near centromeres, and incorrect packaging or expression of some satellite DNA can have profound negative fitness consequences [34–37]. Repetitive elements and their host genome are in a constant arms race, which need to co-evolve to evade each other's response [44, 45]. Mobilization of TEs can cause insertional mutations [43] and natural selection acts to remove deleterious insertions [46]. Several host-encoded mechanisms have emerged during eukaryotic evolution to suppress TE activity [43, 47]. One defense mechanisms widely employed across eukaryotes is transcriptional silencing of repetitive DNA through heterochromatin formation [31]. A host-encoded defense mechanism that prevents movement of a TE family, however, creates selective pressure on these elements to evolve escape mechanisms to avoid the host defense, forcing the host to evolve further mechanisms to silence TEs. Thus, repetitive elements and their host genome are in a constant arms race, where they need to co-evolve to evade each other's response [44, 45].

The effectiveness of host defense mechanisms to eliminate or silence repetitive elements can vary in space and time. Epigenetic defense mechanisms can vary both on evolutionary time scales, and throughout an individual's life. Significant chromatin structural changes occur during aging, with a loss of heterochromatin commonly observed in old individuals [5, 29]. Disruption of heterochromatin during an individual's life can result in mobilization of previously silenced TEs [9, 48]. The effectiveness of epigenetic silencing mechanisms can also vary across evolutionary time scales. In particular, a novel TE that is invading a naïve host genome can escape host surveillance mechanisms and propagate across the genome, until silencing mechanisms evolve to target that TE [32, 49].

Natural selection to remove TE insertions can also be compromised. In the absence of recombination, natural selection is less efficient, and can result in fixation of deleterious mutations, including the accumulation of TEs [50, 51]. Low or non-recombining regions of eukaryotic genomes such as pericentromeres are often an agglomerate of TEs [24, 46, 52], and Y chromosomes of many species, including Drosophila, are comprised almost entirely of repetitive DNA [52, 53]. In fact, the neo-Y of *D. miranda* has accumulated tens of thousands of TE insertions since it stopped recombining 1.5MY ago [14, 21], demonstrating just how quickly TEs can invade a non-recombining chromosome. Non-recombining Y chromosomes of many species can thus act as reservoirs for TEs [53, 54], and dramatically increase the repeat content in males compared to females [12].

The accumulation of deleterious mutations and repetitive elements on the Y chromosome might lower the survival of the heterogametic sex ("toxic Y" hypothesis) [3, 4]. While TEs may be efficiently silenced in young individuals, deteriorating heterochromatin can affect males disproportionally, and result in shorter lifespans in males [12]. Under the toxic Y model, we expect males to show higher levels of TE expression than females, at least during some stages

in their life (for example, in old males when heterochromatin is known to be compromised). Secondly, we would also expect that larger Y chromosomes that contain more repetitive elements are more 'toxic' to males than Y chromosomes with fewer or less active repeats.

Our data support both predictions of the toxic Y effect. We show that repeats from the Y are mis-expressed in male *D. miranda*. We find that the overall number of TE-derived transcripts is about twice as much in males than it is in females, and that TEs located on the Y are primarily responsible for increased TE transcript abundance in males. Secondly, comparison of our data to a recent analysis in *D. melanogaster* suggests that the larger neo-Y of *D. miranda* is a much bigger burden for males than the ancient Y of *D. melanogaster* [12]. Young *D. melanogaster* males are effectively suppressing their Y-linked repeats and epigenetic defense mechanisms are only breaking down in old males [12]. In contrast, TE suppression is compromised even in young *D. miranda* males. The Y chromosome of *D. melanogaster* is substantially smaller than the neo-Y of *D. miranda* and fully heterochromatic [53, 54]. About ¼ of the 40 Mb large Y chromosome of *D. melanogaster* is composed of TEs (that is, about 10 Mb), while the rest of the Y mainly consists of satellite DNA [13]. In contrast, almost 75 Mb of Y/neo-Y derived sequence in *D. miranda* is classified as TE-derived.

In addition to having substantially more repeats, the neo-Y shows lower levels of heterochromatin enrichment compared to repetitive regions on the X or autosomes. The neo-Y of *D. miranda* contains 1000s of expressed genes, and we find that active transcription of these genes appears to impede the formation of silencing heterochromatin at adjacent TEs on the neo-Y. Thus, not only do male *D. miranda* have more repeats than females and their Y-chromosome has more TEs than other species (such as *D. melanogaster*), the TEs present on the neo-Y are particularly poorly silenced, due to the large number of active genes present. Therefore, while Y chromosomes can generally be toxic, the recently formed neo-Y of *D. miranda* is especially harmful [55]. This increased toxicity of the *D. miranda* neo-Y can explain why Y-linked TEs are ineffectively silenced throughout the life of a fly, and not only in old males. Note, however, that our comparative study does not allow us to directly demonstrate that the neo-Y of *D. miranda* reduces male lifespan. Careful genetic manipulations of the sex chromosomes, as done in *D. melanogaster* [12], are necessary to link the absence or presence of the neo-Y to changes in lifespan.

Overall, we detect higher levels of expression of TEs in male flies relative to female flies, both for young and old flies. Our analysis suggests that low levels of heterochromatin on the neo-Y may be inefficient to fully suppress repetitive elements on this chromosome even in young males. Genome-wide deterioration of heterochromatin in old males may further decrease TE silencing, but may not substantially contribute to overall TE transcript abundance in old males that already have a large, inefficiently silenced highly repeat-rich neo-Y. Indeed, this resembles our findings in *D. melanogaster* flies with additional Y chromosomes [12]. While wildtype *D. melanogaster* males efficiently suppress their Y-linked repeats when young, flies with additional Y chromosomes (XXY females or XYY males) show inefficient silencing of Y-linked repeats already in young flies [12]. These patterns are consistent with a dynamic view of co-evolution between selfish TEs and host defense mechanisms. Flies typically can suppress wildtype levels of TE activity when young, but these defense mechanisms may fail as flies age (especially in males which have more repetitive DNA) or when they are being challenged with unusually high numbers of active TEs (extra Y chromosomes in XXY and XYY *D. melanogaster*, or the recently formed, highly repetitive neo-Y of *D. miranda*).

The molecular mechanisms underlying increased TE expression during aging, and the contribution of TE activity to aging are only beginning to be understood. A direct link between neurodegeneration and heterochromatin was found in Drosophila: The microtubule-associated protein tau is involved in a number of neurodegenerative disorders, including

Alzheimer's disease. Wide-spread heterochromatin loss was found in tau transgenic *Drosophila* and mice, and oxidative stress and subsequent DNA damage were identified as mechanistic links between transgenic tau expression and heterochromatin deterioration [56]. Several TEs become highly active in Drosophila brain during normal aging, and exacerbated TE expression in brain results in age-dependent memory impairment and shortened lifespan [11]. Movements of TEs result in DNA damage, and may provide the mechanistic link between TE activity and physiological aging. In mouse, L1 retrotransposons become active during the course of aging; SIRT6—a powerful repressor of L1 activity—binds to the 5'UTR of L1 elements, and facilitates heterochromatin formation [57]. During aging, and also in response to DNA damage, SIRT6 becomes depleted from L1, allowing the activation of these previously silenced TEs [57]. De-repression of L1 elements results in activation of the type-I interferon response and age-associated inflammation, a phenotype of late senescence [58]. A direct link between systemic inflammation and heterochromatin was also observed in Drosophila: genes involved in immune response are enriched in lamin-associated heterochromatin in both flies and mammals, and fat body immunosenecence is caused by age-associated lamin-B reduction, contributing to loss of heterochromatin and de-repression of genes involved in immune responses [59]. Future research will further establish the mechanistic connections between repetitive DNA, TE activity, heterochromatin loss, and sex-specific aging.

## Materials & methods

### Drosophila strains

We used the reference genome strain MSH22 for analysis [14]. For chromatin and gene expression analyses, flies were grown in incubators at 18˚C, 60% relative humidity, and 12h light for the indicated number of days following eclosion, and were then flash-frozen in dry ice and stored at -80˚C. Flies were reared on standard molasses fly food: 0.68% agar (w/v), 2.7% yeast (w/v), 6.67% cornmeal (w/v), 0.456% propionic acid (v/v), 1.6% sucrose (w/v), 0.76% of 95% ethanol (v/v), 0.09% Tegosept (w/v), 8.2% molasses (v/v), 0.0625% CaCl2 (w/v), 0.75% Na Tartrate (w/v).

### Lifespan assays and tissue preparation

Lifespan assays were conducted following {Linford, 2013 #2008} using the same rearing conditions as described above. Briefly, we collected synchronized embryos on a molasses plate with yeast paste for 16–20 hours. We washed embryos with 1x PBS pH 7.4 three times and dispensed 10μl of embryos per culture vial by pipette. To obtain synchronized adult flies, we collected emerging adults over a 3 day window and aged the adults for at least 7 days to mature and copulate. Afterwards, we separated males and females into separate vials, placing 20 individuals per vial, maintaining them at 18˚C with a light:dark setting of 12h:12h. We moved flies to new vials every 3–5 days, without using $CO_2$, and fly deaths were recorded. Flies that were observed escaping the vial were censored. To collect samples for the RNA-seq and ChIP-seq experiment, we censored the entire lifespan experiment once it reached less than 50% survivorship for one sex and flash-froze the remaining flies in liquid nitrogen. In total, 1765 female flies in 107 vials and 1304 male flies in 75 vials were counted for the lifespan assays reported here. We dissected brains from males and females aged 5–9 days (young flies) and 80–98 days old in 1X PBS pH 7.4 (1 brain/sample). We then snap-froze and stored the tissue at -80˚C prior to use in the ChIP or RNA-seq assay. For the spike-in, we dissected brains from *D. melanogaster yw* strain females and pooled 2 brains/sample.

## Chromatin immunoprecipitation and sequencing

We modified the native and ultra-low input ChIP protocol [60] for single-brain ChIP-Seq. First, we added 40μL EZ nuclei lysis buffer to frozen tissues and homogenized them with a pestle grinder on ice. We spun cells down at 1000xg for approximately 10 minutes and decanted 20μL of the supernatant. We froze the cell pellet in the lysis buffer at -80˚C. We then followed the 100,000 cell count digestion protocol from [60] with a few modifications. For each aged *D. miranda* sample and *D. melanogaster* spike-in sample, we added 4.4μL of 1% DOC and 1% Triton-X100 to resuspended cells and mixed thoroughly. Briefly, we digested each sample at 37C for 04:58 (mm:ss) with MNase and quenched the reaction with 4.9μL of 100mM EDTA pH 8.0. We then added the fragmented spike-in samples to each *D. miranda* sample to account for approximately 20% of the final pooled sample volume (i.e. final sample consisted of 80% *D. miranda*, 20% *D. melanogaster*). We reserved 10% of the pooled fragmented sample for the input (fragment control) and used the remaining 90% to perform the chromatin pull-down (ChIP sample, target antibody: H3K9me3 polyclonal classic Diagenode C15410056) according to the 100,000 cell specifications. We used the Rubicon Genomics ThruPlex kit to prepare ChIP-Seq libraries for sequencing with 10 PCR amplification cycles for the input samples and 12 cycles for the ChIP pull-down samples. We sequenced libraries on the Illumina HiSeq 4000 Platform at the Vincent J. Coates Genomics Sequencing Center (Berkeley, CA). In total, we made 4 biological replicates: 2 replicates from the first aging trial, 1 replicate from the second, and 1 replicate from the third.

## RNA extraction and RNA-seq

We extracted total RNA from 20 brains per sex per age (3 biological replicas, one from each aging trial) following a standard TRIzol protocol. Briefly, we homogenized tissues first in 100μl TRIzol and then added 900μl TRIzol to proceed with the full-volume protocol. We generated libraries from total RNA using Illumina's TruSeq Stranded Total RNA Library Prep kit with Ribo-Zero ribosomal RNA reduction chemistry, which depletes the highly abundant ribosomal RNA transcripts (Illumina RS-122-2201). Libraries were run on a Bioanalyzer for fragment traces and sequenced on the Illumina HiSeq 4000 at the Vincent J Coates Genomics Sequencing Center.

## Mapping of sequencing reads and data normalization

We mapped all ChIP-seq datasets to the *D. miranda* genome assembly [14] to the *D. melanogaster* Release 6 of the genome assembly and annotation [61]. For all ChIP-seq datasets, we used Bowtie2 [62] to map reads to both genomes separately, using the parameters "-D 15 –R 2 –N 0 –L 22 –i S,1,0.50—no-1-mm-upfront", which allowed us to reduce cross-mapping to the *D. melanogaster* genome to approximately 1% of 100bp-paired end reads [63]. All ChIP and input samples were processed with bedtools coverage–counts [64]. To calculate the ChIP signal without the *D. melanogaster* spike-in normalization, we normalized all counts files by library size, and calculated the coverage across 5kb windows for both the ChIP and the input *D. miranda* alignments. These values were then normalized by the median autosome coverage of euchromatic arms (Muller B and Muller E) to obtain ChIP signal (ChIP/Input) for each sample. To calculate signals normalized by the *D. melanogaster* spike-in, we used three different approaches (see **S1 Fig**). First, following the approach from [63], we took the ChIP enrichment from the no-spike normalization method and multiplied these values by a scale factor. The scale factor *c* is calculated as follows with normal script indicating the number of reads mapped for the given sequencing sample and subscripts indicating the specific species to which reads were mapped:

$c = 1 + \text{ChIP}_{D.miranda}/(\text{Input}_{D.\ melanogaster}/\text{Input}_{D.\ miranda})/(\text{Input}_{D.\ melanogaster} + \text{ChIP}_{D.\ melanogaster})$. In this respect, we calculated normalization factor $c$ based on the number of mapped reads and this is irrespective of mapping location. We also used a regression-based method from [65]. Briefly, we calculated the median ChIP signal for each sample's *D. melanogaster* spike-ins. Then, the ChIP signal in each *D. miranda* sample was adjusted by a logistic regression based on each sample's genome-wide *D. melanogaster* spike-in signal. In a separate spike-in normalization approach, we used a quantile-quantile method [55]. A reference ChIP signal for all *D. melanogaster* spike-ins was made by aggregating all *D. melanogaster* ChIP and input coverages, respectively. Then, all individual *D. melanogaster* spike-ins were adjusted to this reference and the corresponding *D. miranda* ChIP signals were adjusted accordingly. We emphasize that all spike-in normalization methods do not use specific regions of the genome and are based only on the mapping and/or ChIP enrichment of the *D. melanogaster* spike-in samples. All three spike-in normalization methods as well as the no-spike normalization procedure yielded quantitatively similar results: young males and females show similar enrichment of H3K9me3 along their pericentromeres; both old males and females show less H3K9me3 enrichment than young flies, and the loss of heterochromatin is more pronounced in males than females, and finally, the neo-Y chromosome generally shows less heterochromatin than pericentromeres (**S1 Fig**). We further confirmed the consistency of these normalizations by showing that 5kb windows in euchromatin with elevated H3K9me3 enrichment are found consistently in all normalization methods (**S10 Table**). We used the no-spike normalization procedure for our analysis, and one sample with low ChIP efficacy (old male 1; see **S1 Fig**) was excluded from the analysis.

The boundaries of euchromatic chromosome arms versus pericentromeric heterochromatin was defined using repeat-enrichment, using a cut-off of 40% repeat-masked DNA across 100 kb sliding windows away from the centromere (from [24]. For the neo-Y chromosome, we identified the pericentromeric region based on increased H3K9me3 signal surrounding the centromeric satellite sequences (see [14]), which roughly corresponds to 9.8Mb of sequences at the end of neo-Y scaffold 1 (**S19 Fig**).

## Gene expression analysis

For each replicate of RNA-seq data, we first mapped RNA-seq reads to a repository of ribosomal DNA scaffolds from NCBI and removed all reads that mapped to this scaffold. Although total RNA library preparation aims to remove the bulk of rRNA transcript, any additional differences in rRNA abundance in sequenced samples are likely to be technical artifacts. We then mapped the remaining reads to the *D. miranda* genome and to the repeat libraries separately using HISAT2 [66] with default parameters. We then used Subread FeatureCounts [67] to calculate gene counts and repeat counts and used DESeq2 [68] to normalize the libraries and perform differential expression analysis between male/female flies of different ages. In an analogous approach, we used TECount from TEtranscripts [33] to estimate gene and TE expression counts across samples. We used TECounts with default parameters on RNA-seq reads mapped to the *D. miranda* genome. We also used DESeq2 [68] to perform the differential expression analysis with these counts tables. Both counting methods for repeats yielded similar results for each sample.

For a non-reference based approach, we used dnaPipeTE [69] on RNA-seq reads using the following parameters: -coverage 0.5 -genome-size 200000000 -sample-number 2. We then calculated the proportion of TEs and repeats identified in each sample.

## Repeat libraries

We used two repeat libraries for the ChIP-seq and RNA-seq analyses. Our first library was the consensus sequences of known repetitive elements identified in the *D. pseudoobscura* group

[70] and the second was based on the consensus sequences of known satellites identified in *D. miranda* [14]. The following methods were employed for both repeat libraries.

## ChIP enrichment/RNA expression of specific repeats (and by chromosome)

To assess H3K9me2 signal in repetitive elements, we took a similar approach as we did for calculating ChIP enrichment profiles across the genome. First, we mapped both ChIP and input sequencing reads to the *D. pseudoobscura* family TE library/satellite DNA library using Bowtie2 [62] and the parameters "-D 15 –R 2 –N 0 –L 22 –i S,1,0.50—no-1-mm-upfront". We then calculated the mean coverage across each repetitive element using Bedtools [64] coverage, and normalized the coverage by the median autosome coverage. We then calculated the ChIP signal at each repeat by dividing the ChIP coverage by the corresponding input coverage. This method accounts for differences in copy number of the repetitive elements by dividing the ChIP coverage by each repeat's coverage in the input. In a complementary approach, we took ChIP mappings to the entire genome and called enrichment at repetitive elements through TECounts (part of TETranscripts) [33]. To call for repeat enrichment on a per chromosome basis, we split mappings by chromosome and ran TECounts separately for each split mapping. We then normalized counts at repeats in both the ChIP and input by mean autosomal coverage and calculated ChIP enrichment as ChIP/Input.

## ChIP profiling at transcription start sites (TSS)

To calculate H3K9me3 enrichment around genes on the neo-Y or pericentromeric regions, we followed the BedTools pipeline for estimating transcription factor occupancy at transcription start sites [64]. Briefly, we calculated genome-wide ChIP enrichment at 100bp resolution as described previously. We then generated intervals flanking transcription start sites of genes on the neo-Y and pericentromeric regions using BedTools slop. We then used Bedtools map to map our ChIP enrichments to these flanking regions and Bedtools groupby to aggregate enrichment profiles across all TSS.

## Immunofluorescence staining in mitotic chromosome spreads

We followed an immunofluorescence protocol from [71]. We dissected brains from male 3rd instar larvae in 0.7% NaCl. We incubated brains first in 0.5% sodium citrate for 8 minutes and then in 2% paraformaldehyde in 45% acetic acid fixative solution for 8 minutes. We transferred 4 brains to small drops of fixative solution on a 18mmx18mm siliconized coverslip and placed a clean glass slide on top. The slides were pressed with a thumb for 1.5 minutes and then frozen in liquid nitrogen until bubbling stopped. We used a razor blade to pop off the coverslip and immediately immersed slides in PBS (pH 7.4). The slides were then treated with 100uL of 20units/mL DNAse1 (Thermofisher #89836) in 50mM Tris-HCl pH 8.0, 5mM MgCl2, and 0.05mg/mL BSA for 4.5 minutes. Treatment with DNAse1 increases antibody binding to the H3K9me3 histone mark as reported in [71]. The slides were then washed 3x in PBS for 5 minutes/wash and incubated in 1% Triton X-100 in PBS for 10 minutes. Slides were blocked in 5% nonfat dry milk in PBS for 30 minutes, washed in PBS for 5 minutes, and then incubated with primary antibody (1:300 dilution, Abcam ab8898) in 3% BSA in PBS for 1 hour first at room temperature and later overnight at 4˚C in a box lined with wet paper towels. We washed slides 3x in PBS for 5 minutes/wash and then incubated them with secondary antibody Goat anti-rabbit conjugated with Alexa Fluor-568 in 3% BSA in PBS (1:1000 dilution, Thermofisher #A-11011) for 1 hour at room temperature in a box lined with wet paper towels and covered with aluminum foil. Slides were washed 3x in PBS for 5 minutes/wash and dried for 10 minutes in the dark. We mounted slides using Vectashield antifade mounting medium with

DAPI (Vector Laboratories H-1200-10) on a coverslip and then sealed the edges with nail polish. Slides were imaged using a Nikon Spinning Disc Confocal microscope and the Nikon Elements AR software at the UC Berkeley Cancer Research Laboratory Molecular Imaging Center. We used Fiji/ImageJ to adjust image brightness and contrast.

## Supporting information

**S1 Fig. ChIP enrichment across samples using four different normalization strategies (5kb windows; see Materials & Methods for details).** From top to bottom: No spike in normalization: ChIP enrichment normalized by coverage of the autosomes for all biological samples and replicates, ignoring the spike-in. Brown et al. (2020) normalization: ChIP enrichment normalized by assessing proportion of reads derived from sample and spike-in both the ChIP and Input. Bonhoure et al. (2014) normalization: ChIP enrichment normalized by employing a regression-based method using the *D. melanogaster* spike-in. Wei et al. (2020) normalization: ChIP enrichment normalized by employing a quantile-quantile method using the *D. melanogaster* spike-in. **A.** Enrichment for each sample replicate. **B.** Average enrichment across samples. We omitted the first Old Male sample for all averaged Old Male values because it had lower enrichment signal compared to other samples in both the sample and the spike-in, indicative of inefficient ChIP pull-down.
(PDF)

**S2 Fig. ChIP signal after subsampling reads to 37 million reads for each replicate. A.** Enrichment for each sample replicate. **B.** Average enrichment across samples. Enrichment is normalized without spike-in and omits the failed Old Male replicate.
(PDF)

**S3 Fig. ChIP enrichment across samples (average across replicates) with only uniquely mapped reads (5kb windows) using ChIP signal normalized by autosome coverage (3 replicates used for the Old Male category).**
(PDF)

**S4 Fig. Input coverage across sexes and ages.** (Top) Coverage of the input DNA across samples normalized by the median autosomal coverage. (Bottom) Mean coverage of the input DNA across samples.
(PDF)

**S5 Fig. Genome-wide ChIP enrichment with ± 1 standard error (denoted in yellow).** 1 tick mark = 1Mb on the x-axis.
(PDF)

**S6 Fig. Differential ChIP enrichment between the sexes.** Each data point represents enrichment of old/young male (left) or female (right) samples at a 5—kb window. Data highlighted in yellow denote windows with 50% higher/lower enrichment between samples (p < 0.05, two-tailed Student's t-test).
(PDF)

**S7 Fig. Regions with at least 1.5x H3K9me3 up-/down-regulation within sexes over aging in 5kb-window resolution.** Data values represent the difference of means from old and young samples. One tick mark on the x-axis is equal to 2Mb.
(PDF)

**S8 Fig. Enrichment of H3K9me3 (in 5kb windows) at the pericentromeres and 2Mb downstream for A.** males and **B.** females. Boundaries are indicated by darker region on the

chromosome diagrams of the major chromosome arms. Subtraction plots show higher H3K9me2 signal in young (blue) or old (red) flies. Data represents the mean of 4 biological replicates (3 for old males; in black) or the difference of means (subtraction plots, blue or red). (PDF)

**S9 Fig. Sex-specific H3K9me3 gain/loss by compartment (euchromatin/heterochromatin/ neo-Y) over aging.**
(PDF)

**S10 Fig. Immunofluorescence staining for H3K9me3 in males (A-I, replicates from different slides) and females (J).** For all figures, arrowhead denotes neo-Y, arrows denote Muller-A-AD, DAPI/DNA is blue channel, H3K9me3 is orange channel, and scale bar is 50μm. (PDF)

**S11 Fig. Repeat composition per chromosome.** Plotted for each Muller element is the total length of repeats (bp) divided by the entire chromosome length (bp). (PDF)

**S12 Fig. ChIP enrichment differences between young and old females and males.** Values represent the difference of means between young and old samples, grouped by chromosome and binned by amount of repeats (%) per 5kb-window. (PDF)

**S13 Fig. Volcano plots for TE expression by sex and age.** Left: (Top) TEs in young females v. young males, (bottom) TEs in old females vs old males. Fold change is indicative of male/ female. Right: (Top) TEs in young males vs old males, (bottom) TEs in young females vs old females. Fold change is indicative of young/old. (PDF)

**S14 Fig. TE expression by sex/age and by chromosome group (Autosomes/X and Neo-Y).** (PDF)

**S15 Fig. Male TE expression from the Autosomes/X and the Neo-Y in young (left) and old males (right).** Diagonal line denotes 1:1 equal expression for TEs from both chromosome groups. (PDF)

**S16 Fig. Sex-differences in TE expression normalized by sex-differences in TE copy number. A-C.** Shown are expression values for each biological sample accounting for TE copy number differences in the males and females when copies are calculated from **A.** adult gDNA, **B.** embryo gDNA, and **C.** genome repeat annotation. Expression values for each TE were normalized by their respective copy number in the male or female genome. Data plotted represents the averaged values of old and young female and male samples (3 replicates). Significance values calculated (n.s. not significant, $^{*}$ p<0.05, Wilcoxon test). **D-F.** Shown are Log2(Male/Female Expression) values for each TE along with the corresponding Log2(Male/ Female DNA) values calculated from **D.** adult gDNA, **E.** embryo gDNA, and **F.** genome repeat annotation. Line indicates male/female expression is proportional to the male/female copy number of a specific TE. (PDF)

**S17 Fig. Volcano plots for satellite DNA expression by sex and age.** Left: (Top) Satellites in young females v. young males, (bottom) satellites in old females vs old males. Fold change is indicative of male/female. Right: (Top) Satellites in young males vs old males, (bottom)

satellites young females vs old females. Fold change is indicative of young/old.
(PDF)

**S18 Fig. Heterochromatin and expression of pericentromeric and neo-Y genes. A.** Log2 expression of genes located in the pericentromeres within sexes. Data represents mean values of 3 replicates with bars denoting standard error. Data highlighted in yellow denotes genes with significantly 50% higher or lower expression when comparing fold-change between samples (Wald test, p < 0.05). Grey line indicates 1:1 expression between samples. Boxplot values represent mean values of 3 replicates with significance values calculated using the Wilcoxon test (* p< 0.05, ** p<0.01, *** p<1e-5). **B.** Log2 expression of genes located on the neo-Y between young and old males. Data represented in a similar manner as in **A** and data highlighted shows genes that are up-regulated (top-left) and/or down-regulated (bottom-right) by 50% during aging (Wald test, p < 0.05). **C.** Age-related differences in log2 H3K9me3 enrichment and log2 expression of neo-Y genes. Each data represents the difference of mean values between young and old males. Color-code indicates differentially expressed/enriched comparisons: black: absolute log2(fold-change) < 1in both H3K9me3 and expression; yellow: absolute log2(fold-change)> = 1 in both H3K9me3 and expression. **D.** Log2 expression differences in genes on the neo-Y and their respective gene copy number. Values denote the difference of mean values between young and old males. Grey line denotes 0 expression difference between the ages.
(PDF)

**S19 Fig.** Heterochromatin cutoffs (dashed orange line) by % repeats overlap with regions of elevated H3K9me3 enrichment on (A) Muller B, (B) Muller E, (C) Muller C, and (D) Muller AD.
(PDF)

**S1 Table. Overview of ChIP data generated.** Description of samples/tissues used for 4 rounds/batches of ChIP-seq experiments.
(PDF)

**S2 Table. Mapping statistics of ChIP data.** Description of mapping statistics for ChIP-experiment data for each batch. Reported numbers of reads mapped to the sample and spike-in genomes.
(PDF)

**S3 Table. Pearson coefficients between replicates of same biological sample.** We used H3K9me3 enrichment in 5kb windows from the autosomes and X chromosomes to conduct Pearson correlations between each pair of replicates. Pearson correlations were calculated using the base R function cor.test(replicateA, replicateB, method ="pearson").
(PDF)

**S4 Table. Proportion of 5kb-windows in top 20 percentile H3K9me3 enrichment shared between normalization methods.** For each normalization method, we identified the top 20% most enriched 5kb windows in euchromatin (N = 4646). We then identified how many 5kb regions were shared between each normalization method and reported those results as a percentage (no. windows shared between two methods / no. windows in top 20% enrichment).
(PDF)

**S5 Table. Mapping statistics of ChIP data for MAPQ > 3 alignments.**
(PDF)

**S6 Table. Heterochromatin-related genes and their expression change in old males and female.**
(PDF)

**S7 Table. Overview of RNA data generated.** Description of samples/tissues used for 3 rounds/batches of RNA—seq experiments.
(PDF)

**S8 Table. Mapping and counts statistics of RNA data.** Description of mapping statistics for RNA-seq experiment data for each batch. Reported gene and repeat counts by Subread feature-counts and TEtranscripts where applicable.
(PDF)

**S9 Table. Mapping and counts statistics of RNA data filtered for MAPQ > 40.**
(PDF)

**S10 Table. DNAPipeTE estimates of repeats in RNASeq data.**
(PDF)

**S1 Appendix.** *D. melanogaster* **as a spike-in to normalize** *D. miranda* **ChIP's.**
(PDF)

**S2 Appendix. Cross-mapping between the** *D. miranda* **neo-X and neo-Y chromosome.**
(PDF)

## Acknowledgments

We thank Laura Fanti for help with the immunostaining protocol.

## Author Contributions

**Conceptualization:** Alison H. Nguyen, Doris Bachtrog.

**Data curation:** Alison H. Nguyen.

**Formal analysis:** Alison H. Nguyen.

**Funding acquisition:** Doris Bachtrog.

**Investigation:** Alison H. Nguyen, Doris Bachtrog.

**Methodology:** Alison H. Nguyen, Doris Bachtrog.

**Project administration:** Doris Bachtrog.

**Resources:** Alison H. Nguyen, Doris Bachtrog.

**Software:** Alison H. Nguyen.

**Supervision:** Doris Bachtrog.

**Validation:** Alison H. Nguyen.

**Visualization:** Alison H. Nguyen, Doris Bachtrog.

**Writing – original draft:** Doris Bachtrog.

**Writing – review & editing:** Alison H. Nguyen, Doris Bachtrog.

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
