## [Decision Letter · Decision Letter 0]

27 Oct 2020

Dear Dr Bachtrog,

Thank you very much for submitting your Research Article entitled 'Toxic Y chromosome: increased repeat expression and age-associated heterochromatin loss in male Drosophila with a young Y chromosome' to PLOS Genetics. Your manuscript was fully evaluated at the editorial level and by independent peer reviewers. The reviewers appreciated the attention to an important problem, but raised some substantial concerns about the current manuscript. The reviers' most significant concerns appear to be:

Two issues need to be fully addressed before this study can be accepted for publication in PLoS Gen.

1. Orthogonal validation that H3K9me3 is indeed lower in the neo-Y chromosome

2. Detailed description and demonstrated robustness of their analysis with respect to repetitive sequences.

These will ensure that this study's two major findings are true (neo-Y has lower H3K9me3 enrichment and TEs are de-repressed in young males already). 

There are also numerous concerns detailed in the two reviews.  While we are very aware  of this limits the pandemic places on doing additional experiments, we do ask that you do the experiments you can and repeat and detail the analyses and writing as fully as possible

We hope to be be able to review again a much-revised version.

Your evisions should address the specific points made by each reviewer. We will also require a detailed list of your responses to the review comments and a description of the changes you have made in the manuscript.

If possible, please aim to resubmit within the next 60 days, unless it will take extra time to address the concerns of the reviewers, in which case we would appreciate an expected resubmission date by email to plosgenetics@plos.org.

[LINK]

We are sorry that we cannot be more positive about your manuscript at this stage. Please do not hesitate to contact us if you have any concerns or questions.

Yours sincerely,

R. Scott Hawley

Associate Editor

PLOS Genetics

Kirsten Bomblies

Section Editor: Evolution

PLOS Genetics

Reviewer's Responses to Questions

**Comments to the Authors:**

Reviewer #1: In this study, Nguyen and Bachtrog addressed an outstanding question in evolutionary genomics – how does the evolution of sex chromosome influence epigenomes and, accordingly, individual phenotypes ("aging"). Studies from the Bachtrog lab previously reported that, compared to females, silenced transposable elements (TEs) and satellite repeats are derepressed in aged D. melanogaster males due to the "toxic Y" effect. In this study, authors reported that in a species with recently evolved neo-Y chromosome, D. miranda, there is similar derepression of TEs and satellite repeats in males. However, the derepression of TEs is not restricted to aged males, but also present in young males. Authors suggested that the much larger neo-Y chromosome is a bigger "burden" and thus more toxic. Another major finding from the study is that, compared to other heterochromatic regions, neo-Y has lower enrichment of heterochromatic marks, and the loss of heterochromatin differs between males and females. While these findings are exciting and could provide new insights for the field, there are several potential caveats that need to be formally addressed to ensure the robustness of the authors' findings. Below I listed my specific comments.

Major issues

1. Most of the major conclusions from this study were drawn from the analysis of H3K9me3 enrichment and the expression of repetitive sequences (satellite repeats or TEs). Because of their repetitive nature, reads from these sequences will appear as "multi-mapped reads" in ChIP-seq or RNA-seq data. However, the authors did not provide enough details about how they address the multiple-mapping issue.

Some of the reported analyses made me especially confused. For instance, in page 6, the authors stated that the results are consistent when using 5kb windows with different percentages of repetitive sequences, even for windows with >99.5% repetitive sequences. This makes it sound like that the authors only focused on unique sequences when estimating H3K9me3 enrichment (but also see below). However, 0.005*5,000 = 25, meaning that the H3K9me3 estimates were based on merely 25bp of unique sequence?

If authors only used unique sequences, it is unclear what the filtering criteria are (e.g., mapping quality). If authors allowed multi-mapped reads, then how did bowtie assign multi-mapped reads, or how authors decided where a repetitive read came from?

The authors also compared the H3K9me3 enrichment level between TEs on the neo-Y and other pericentromeric heterochromatin. How did the authors distinguish repetitive TEs in different genomic locations? Did they use the H3K9me3 enrichment in the unique flanking sequence (if so, data need to be shown)?

In addition, how the RNA-seq comparison is normalized with respect to TE copy number is not clear. Especially, there are TE copies inside and outside heterochromatin regions. How the authors distinguished repetitive copies within and outside heterochromatin (and even more, on neo-Y and other chromosomes)?

A related question - the authors used D. melanogaster spike-in as a control to normalize between samples. While there is enough genetic divergence in unique sequences between the two species, it is unclear whether repetitive sequences also diverge enough to avoid cross-mapping. Could it be possible that D. melanogaster repeats from the spike-in confound the estimates for D. miranda?

2. One major finding from this study is that, compared to pericentromeric heterochromatin of other chromosomes, neo-Y has lower enrichment of H3K9me. Several aspects of this finding need to be further addressed.

- How non-unique sequences (not necessarily repetitive) were treated in the analysis? I presume that the neo-Y still has many shared sequences with the neo-X. A neo-Y ChIP-seq read could also map to neo-X, leading to averaging out the H3K9me3 enrichment level of estimated for neo-Y.

- Because of the uncertainty of how repetitive (#1) and non-unique (see above) sequences are treated in the reported ChIP-seq analysis, I feel it is necessary to use orthogonal methods to demonstrate that neo-Y indeed has lower enrichment of H3K9me3. A straightforward IF experiment should be able to address this.

- Authors suggested that the lower enrichment of H3K9m3 on the neo-Y could be due to the active transcription of genes on the neo-Y. This possibility could and should be further investigated, potentially with existing data. For example, authors can compare the H3K9me3 enrichment level in windows with different distances to transcribing genes, genes with different transcription levels etc, for both neo-Y and heterochromatin on other chromosomes.

- A related and important question - it is intriguing how authors defined heterochromatin differs between neo-Y (using H3K9me3 gradient) and other chromosomes (using % of repeats). Heterochromatin is genomic regions enriched with H3K9me3. I am not sure why the authors chose other metrics for chromosomes other than the neo-Y. I also wonder whether some results could have been driven by this difference in methods between neo-Y and other heterochromatin.

Other comments

3. Authors should include line numbers – it is hard to refer to specific passages without line numbers.

4. In order to know that authors' normalization methods work, it would be helpful to show some euchromatic loci/genes that are known to be stably enriched for K9. Especially, I am concerned about the cross-mapping of D. melanogaster sequences from the spike-in (#1).

A related question – Did the authors use orthologous euchromatic regions? Or any euchromatic regions? Only unique reads? If allowing any euchromatic regions and could be multi-mapped reads, I am worrying that the normalization could be influenced by the difference in H3K9me3 level on the neo-Y between samples (since neo-X and neo-Y still share some sequences, also see #2).

5. I am confused with the authors' argument that active TE families are less epigenetically silenced, and this could explain the lower K9 enrichment on the neo-Y (page 6). At least in D. melanogaster, LTR families are recently active, targeted by a large amount of small RNAs, and more epigenetically silenced.

6. Authors need to explain why the epigenetic silencing of simple repeats and TEs in young and old males differ in D. miranda and D. melanogaster. The toxic Y hypothesis suggests that the source/sink relationship is the key, but does not distinguish between repeat types. This is central to the main theme of this study and needs to be discussed in depth.

Related to this - discussion on page 12 is confusing.

Authors started with that repeats on the Y lead to the toxic effect, but then stated that the differences between D. melanogaster and D. simulans are due to their difference in TE content on the Y. Authors failed to explain why the different TE content matters and whether the toxic Y theory is relevant only for TEs or both repeats.

7. (Page 16) Authors excluded one old male sample (old male 1) based on the boxplot in Figure S1. However, the criterion for excluding that sample is unclear. By eyeballing, I feel that young female 2 is an outlier to other replicates and could also be excluded. Authors may consider more formal analysis (e.g., IDR) to include/exclude a replicate.

8. (Page 5) Authors reported that the majority of the windows that lost K9 are in pericentromeric heterochromatin or neo-Y. However, these genomic regions are also enriched with K9-enriched windows. It could be that the whole genome is losing K9. A more appropriate analysis would be to compare the proportion of windows in each genomic compartment (Het, neo-Y, Eu) that lost K9.

Related to this - it is intriguing that the number of windows with increased K9 is also more for males than females. I wonder whether this has to do with the uneven coverage between males and females (assuming sequencing depth is the same, the average coverage would be higher for females due to the overall smaller genome size). This could lead to a larger variance for estimates in males. This could potentially be addressed by downsampling female data to the same genome coverage.

9. Authors mentioned that male-male aggression could also lead to lower survival of the males. Since the authors mentioned it, it would be helpful to provide either a rationale why this did not impact the authors' results or some empirical validation (e.g., rear flies individually to estimate the lifespan).

10. In Figure S13, no p-values were provided (neither in the text). By looking at the figure, there are no obvious differences in TE expression between males and females (which contradicts with authors' claim on page 9).

11. I am not sure what H3K9me3 Enrichment "Signal" refers to in Figure 2 and some other figures. I think the authors meant "H3K9me3 Fold Enrichment"?

12. Some of the numberings of supplementary figures are wrong (e.g., Figure S11 on page 9, some of those should be Figure S12).

Reviewer #2: This is a very nice manuscript describing sex-specific derepression of transposable elements with age in a Drosophila species with a large Y chromosome. The manuscript is well-written and clear, though I would like to see some minor points addressed.

An alternative to the toxic-Y explanation for some of the results is a male-specific decline in proteins required for heterochromatin formation. I'd agree that this is less likely, but perhaps they can address this alternative with a few more analyses of the RNAseq and ChIPseq data already collected-- e.g., is there a decline in expression in males with age of any relevant genes that is strong relative to that in females? Are overall H3K9me3 levels similar between males and females of the same age? This may be particularly important to show for X-linked genes, particularly as the neo-X in this group is, as I recall, not fully dosage compensated.

While I appreciate that the authors focussed on well-founded conclusions, I did feel they could speculate a bit more about the proximate causes of the hypothesized toxic Y-effect. e.g., TE activity leading to DNA damage and cell-death in the soma. On a related point, there is something of a lack of non-fly literature on this topic cited (e.g.,de Cecco et al Nature volume 566, pages73–78(2019)), where some of these details have been hammered out.

Similarly, I was surprised to see no mention of the essentially similar work of Lemos et al (e.g., PNAS September 7, 2010 107 (36) 15826-15831)

Finally, I am a wondering how well the attempt to correct expression for TE copy numbers works. The standardisation used, as I understand it, is to divide RNAseq reads by DNA reads. While logical, it is worth remembering that heterochromatin is underreplicated, so that TE copies that are effectively heterochromatin-ised by be underrepresented in the DNA data, and therefore undercounted. I don't think this materially affects their conclusions, but would appreciate an argument showing this, as it is effected by exactly the process they are trying to measure. Or, alternatively, they could correct with early embryo DNAseq data if any exists. (As heterochromatin does not yet form in early embryos.)

p8-- the word 'access' is used, but I think 'excess' is meant.

**Have all data underlying the figures and results presented in the manuscript been provided?**

Reviewer #1: Yes

Reviewer #2: **No: **There is a Genbank Bioproject number in the ms, but a search of Genbank brought nothing up. Presumably the data are currently embargoed, and will be available upon publication.

PLOS authors have the option to publish the peer review history of their article (what does this mean?). If published, this will include your full peer review and any attached files.

Reviewer #1: No

Reviewer #2: No

---

## [Editor Report · Decision Letter 1]

22 Feb 2021

Dear Dr Bachtrog,

We commend the authors for a beautiful Response to Reviewers. The authors so much more than 'did the work'.  The Review Figures were both well done and very helpful.  (Perhaps the authors might wish to add them to the Supplementary Figures?) Either way, this is truly scholarly.  Thank you and congratulations on a terrific paper!

That said, we are pleased to inform you that your manuscript entitled "Toxic Y chromosome: increased repeat expression and age-associated heterochromatin loss in male Drosophila with a young Y chromosome" has been editorially accepted for publication in PLOS Genetics. Congratulations!

Yours sincerely,

R. Scott Hawley

Associate Editor

PLOS Genetics

Kirsten Bomblies

Section Editor: Evolution

PLOS Genetics

Comments from the reviewers (if applicable):

**Data Deposition**

http://datadryad.org/submit?journalID=pgenetics&manu=PGENETICS-D-20-01425R1

**Press Queries**

---

## [Editor Report · Acceptance letter]

26 Mar 2021

PGENETICS-D-20-01425R1 

Toxic Y chromosome: increased repeat expression and age-associated heterochromatin loss in male Drosophila with a young Y chromosome 

Dear Dr Bachtrog, 

We are pleased to inform you that your manuscript entitled "Toxic Y chromosome: increased repeat expression and age-associated heterochromatin loss in male Drosophila with a young Y chromosome" has been formally accepted for publication in PLOS Genetics! Your manuscript is now with our production department and you will be notified of the publication date in due course.

With kind regards,

Katalin Szabo

PLOS Genetics

On behalf of:
